Benchmark datasets for phylogenomic pipeline validation, applications for foodborne pathogen surveillance

Timme Ruth E. ruth.timme@fda.hhs.gov 1
Rand Hugh 1
Shumway Martin 2
Trees Eija K. 3
Simmons Mustafa 4
Agarwala Richa 2
Davis Steven 1
Tillman Glenn E. 4
Defibaugh-Chavez Stephanie 5
Carleton Heather A. 3
Klimke William A. 2
Katz Lee S. 3 6
1 Center for Food Safety and Applied Nutrition, US Food and Drug Administration , College Park , MD , United States of America
2 National Center for Biotechnology Information, National Institutes of Health , Bethesda , MD , United States of America
3 Enteric Diseases Laboratory Branch, Centers for Disease Control and Prevention , Atlanta , GA , United States of America
4 Food Safety and Inspection Service, US Department of Agriculture , Athens , GA , United States of America
5 Food Safety and Inspection Service, US Department of Agriculture , Wahington , D.C. , United States of America
6 Center for Food Safety, College of Agricultural and Environmental Sciences, University of Georgia , Griffin , GA , United States of America
Crandall Keith
Electronic publication date: 2017 Oct 6
Publication date: 2017
Volume: 5
Electronic Location ID: e3893
Received 2017 Jul 21; Accepted 2017 Sep 15
Copyright year: 2017
License: This is an open access article, free of all copyright, made available under the Creative Commons Public Domain Dedication. This work may be freely reproduced, distributed, transmitted, modified, built upon, or otherwise used by anyone for any lawful purpose.
License URL: https://creativecommons.org/publicdomain/zero/1.0/

Keywords: Benchmark datasets, Phylogenomics, Food safety, Foodborne outbreak, Salmonella, Listeria, E. coli, Validation, WGS

Funding: Center for Food Safety and Applied Nutrition at the Food and Drug Administration Advanced Molecular Detection (AMD) Initiative at Centers for Disease Control and Prevention Intramural Research Program of the National Institutes of Health, National Library of Medicine USDA-FSIS program This work was supported by the Center for Food Safety and Applied Nutrition at the Food and Drug Administration; the Advanced Molecular Detection (AMD) Initiative at Centers for Disease Control and Prevention; the Intramural Research Program of the National Institutes of Health, National Library of Medicine; and USDA-FSIS program funding. The funders had no role in study design, data collection and analysis, decision to publish, or preparation of the manuscript.

==============================
Background

As next generation sequence technology has advanced, there have been parallel advances in genome-scale analysis programs for determining evolutionary relationships as proxies for epidemiological relationship in public health. Most new programs skip traditional steps of ortholog determination and multi-gene alignment, instead identifying variants across a set of genomes, then summarizing results in a matrix of single-nucleotide polymorphisms or alleles for standard phylogenetic analysis. However, public health authorities need to document the performance of these methods with appropriate and comprehensive datasets so they can be validated for specific purposes, e.g., outbreak surveillance. Here we propose a set of benchmark datasets to be used for comparison and validation of phylogenomic pipelines.

Methods

We identified four well-documented foodborne pathogen events in which the epidemiology was concordant with routine phylogenomic analyses (reference-based SNP and wgMLST approaches). These are ideal benchmark datasets, as the trees, WGS data, and epidemiological data for each are all in agreement. We have placed these sequence data, sample metadata, and “known” phylogenetic trees in publicly-accessible databases and developed a standard descriptive spreadsheet format describing each dataset. To facilitate easy downloading of these benchmarks, we developed an automated script that uses the standard descriptive spreadsheet format.

Results

Our “outbreak” benchmark datasets represent the four major foodborne bacterial pathogens (Listeria monocytogenes, Salmonella enterica, Escherichia coli, and Campylobacter jejuni) and one simulated dataset where the “known tree” can be accurately called the “true tree”. The downloading script and associated table files are available on GitHub: https://github.com/WGS-standards-and-analysis/datasets.

Discussion

These five benchmark datasets will help standardize comparison of current and future phylogenomic pipelines, and facilitate important cross-institutional collaborations. Our work is part of a global effort to provide collaborative infrastructure for sequence data and analytic tools—we welcome additional benchmark datasets in our recommended format, and, if relevant, we will add these on our GitHub site. Together, these datasets, dataset format, and the underlying GitHub infrastructure present a recommended path for worldwide standardization of phylogenomic pipelines.

Introduction

Foodborne pathogen surveillance in the United States is currently undergoing an important paradigm shift: pulsed-field gel electrophoresis (PFGE) (Swaminathan et al., 2001) is being replaced by the much higher resolution whole genome sequencing (WGS) technology. The data are also more accessible as the raw genome data are now being made public immediately after collection. These advances began with an initial pilot project to build a public genomic reference database, “GenomeTrakr” (Allard et al., 2016) for pathogens from the food supply and has matured through a second pilot project to collect WGS data and share it publicly in real time for every Listeria monocytogenes isolate appearing in the US food supply (both clinical and food/environmental isolates) (Jackson et al., 2016). The Real-Time Listeria Project was initiated by PulseNet, the national subtyping network for foodborne disease surveillance, and is coordinated by the Centers for Disease Control and Prevention (CDC), the Food and Drug Administration (FDA), the National Center for Biotechnology Information (NCBI), and the Food Safety and Inspection Service (FSIS) of the United States Department of Agriculture. The success of the project confirmed that a national laboratory surveillance program using WGS is possible and highly efficient. Now, genome data are collected in real-time for five major bacterial foodborne pathogens (Salmonella enterica, Listeria monocytogenes, Escherichia coli, Vibrio parahaemolyticus and Campylobacter spp.); WGS data are being deposited in either the Sequence Read Archive (SRA) or GenBank, and are being clustered into phylogenetic trees using SNP analysis; results are publicly available at NCBI’s pathogen detection website (NCBI, 2017). The list of pathogens under active genomic surveillance is growing. As of August 16th, 2017, over 150 thousand genomes have been sequenced and contributed towards this public pathogen surveillance effort.

The collaboration among the FDA, NCBI, FSIS, and CDC has been formalized as the Genomics and Food Safety group (Gen-FS) (CDC, 2015). One of the first directives for Gen-FS is ensuring consistency across the different tools for phylogenomic analysis used by group participants. The best way to accomplish this is to have standard benchmark datasets, which enable researchers to assess the consistency of results across different tools and between version updates of any single tool. Each agency has been using compatible bioinformatics workflows for their WGS analysis. PulseNet-participating laboratories use whole genome multilocus sequence typing (wgMLST) with core-genome multilocus sequence typing (cgMLST) at its core (Moura et al., 2016). NCBI uses the Pathogen Detection Pipeline (NCBI, 2017). At the FDA, the Center for Food Safety and Applied Nutrition (CFSAN) uses SNP-Pipeline (Davis et al., 2015). The CDC uses Lyve-SET (Katz et al., 2017). These methods have been designed to match the specific needs of the different agencies performing bacterial foodborne pathogen surveillance. For example, PulseNet surveillance identifies clusters of closely related clinical isolates from cases of foodborne disease that may be followed up in outbreak investigations by all three agencies. After the WGS and epidemiological evidence are considered the FDA and FSIS conduct further investigations and take appropriate regulatory actions. Other phylogenomic analysis packages could also benefit from standardized benchmark datasets, e.g., NASP, Harvest, kSNPv3, REALPHY and SNVPhyl (Gardner & Hall, 2013; Treangen et al., 2014; Bertels et al., 2014; Sahl et al., 2016; Petkau et al., 2017). Consistent validation of the many available analysis packages is essential if we are to use genomic data for regulatory action.

Many pathogen outbreak datasets with raw reads have been made public, for example, genomes from several North American Listeria monocytogenes events (Chen et al., 2016; Chen et al., 2017b; Chen et al., 2017a) a Yersinia pestis outbreak from North America (Sahl et al., 2016), a Clostridioides difficile outbreak dataset from the UK (Treangen et al., 2014), a Clostridioides difficile outbreak in the UK (Eyre et al., 2013), the S. enterica subsp. enterica serovar Bareilly (S. enterica ser Bareilly) 2012 outbreak in the US (Hoffmann et al., 2015), and an S. enterica subsp. enterica serovar Enteritidis outbreak in the UK (Quick et al., 2015). Additionally, many datasets have been published during the course of peer review for this paper, making it difficult to keep track of all datasets. However, they are not in a standardized format, making them difficult to acquire or use in automated analyses. As of September 2017, no bacterial outbreak datasets have been specifically published for use as benchmark datasets. Here we present a set of outbreak benchmark datasets for use in comparison and validation of phylogenomic pipelines.

Materials & Methods

We present one empirical dataset for each of four major foodborne bacterial pathogens (L. monocytogenes, S. enterica ser. Bareilly, E. coli, and C. jejuni) and one simulated dataset generated from the S. enterica ser. Bareilly tree using the pipeline TreeToReads (McTavish et al., 2017), for which both the true tree and SNP positions are known. In addition, we propose a standard spreadsheet format for describing these and future benchmark datasets. That format can be readily applied to any other bacterial organism and supports automated data analyses. Finally, we present Gen-FS Gopher, a script for easily downloading these benchmark datasets. All of these materials are freely available for download at GitHub: https://github.com/WGS-standards-and-analysis/datasets.

Each of the four empirical datasets is either representative of a food recall event in which food was determined to be contaminated with a specific bacterial pathogen, or of an outbreak in which at least three people were infected with the same pathogen. In all four datasets, the results of the epidemiological investigation and the phylogenomic analyses are in concordance. In other words, all isolates implicated in a given event share a common ancestor, or cluster together, in the phylogeny. Although it might be tempting to place these four datasets in the context of a transmission network, it is not the appropriate usage. A phylogeny (with clinical and environmental isolates at the tips and inferred ancestors at internal nodes) is more appropriate due to the nature of foodborne outbreaks: point sources that usually originate from food vehicles, whereas a transmission network more appropriately models person-to-person transmission events. Although our particular four datasets are not intended for transmission network analysis, this does not prevent any future datasets with this intended usage. On the contrary, we have included a field “intendedUse” which addresses this issue and helps future-proof the proposed dataset format (Table 1). All isolates listed in these benchmark datasets were sequenced at our federal or state-partner facilities, using either an Illumina MiSeq (San Diego, CA, USA) or a Pacific Biosciences (PacBio) instrument (Menlo Park, CA, USA).

Table 1 Metadata table header.

Available key/value pairs that describe the entire dataset. Organism and source are required but other key/value pairs are optional.

Key	Description	Example value(s)	
Organism	The genus, species, or other taxonomic description	Listeria monocytogenes	
Outbreak	Usually the PulseNet outbreak code, but any other descriptive word with no spaces	1408MLGX6-3WGS	
PMID	The Pubmed identifier of a related publication	25789745	
Tree	The URL to a newick-formatted tree	http://api.opentreeoflife.org/v2/study/ot_301/tree/tree2.tre	
Source	A person who can be contacted about this dataset	Cheryl Tarr	
DataType	Either empirical or simulated	Empirical	
IntendedUse	Why this dataset might be useful for someone in bioinformatics testing	Epidemiologically and laboratory confirmed outbreak with outgroups	

The simulated dataset was created using the TreeToReads v 0.0.5 (McTavish et al., 2017), which takes as input a tree file (true phylogeny), an anchor genome, and a set of user-defined parameter values. We used the S. enterica ser. Bareilly tree as our “true” phylogeny and the closed reference genome (CFSAN000189, GenBank: GCA_000439415.1) as our anchor. The parameter values were set as follows: number_of_variable_sites = 150, base_genome_name = CFSAN000189, rate_matrix = 0.38, 3.83, 0.51, 0.01, 4.45, 1, freq_matrix = 0.19, 0.30, 0.29, 0.22, coverage = 40, mutation_clustering = ON, percent_clustered = 0.25, exponential_mean = 125, read_length = 250, fragment_size = 500, stdev_frag_size = 120. The output is a pair of raw MiSeq fastq files for each tip (simulated isolate) in the input tree and a VCF file of known SNP locations.

Maximum likelihood phylogenies included for each dataset were inferred by first gathering SNPs from SNP Pipeline (Davis et al., 2015) and then using Garli version 2.01 (Zwickl, 2006) for phylogenetic reconstruction on each resulting SNP matrix.

Results/Discussion

The L. monocytogenes dataset (Table S1) comprises genomes spanning the genetic diversity of the 2014 stone fruit recall (Jackson et al., 2016; Chen et al., 2016). In this event, a company voluntarily recalled certain lots of stone fruits (peaches and the like) based on the company’s internal tests, which were positive for the presence of L. monocytogenes. This dataset describes a polyclonal phylogeny having three major subclades, two of which include clinical cases. The genome for one isolate was closed, yielding a complete reference genome. This dataset also includes three outgroups that were not associated with the outbreak.

The C. jejuni dataset (Table S2) represents a 2008 outbreak in Pennsylvania associated with raw milk (MarlerClark, 2008). This dataset reflects a clonal outbreak lineage with several outgroups not related to the outbreak strain.

The E. coli dataset (Table S3) is from a 2014 outbreak in which raw clover sprouts were identified as the transmission vehicle (CDC, 2014). Nineteen clinical cases had the same clone of Shiga-toxin-producing E. coli O121. The genome for one isolate that was epidemiologically unrelated to the outbreak but phylogenetically related was closed, yielding a complete reference genome. Only three of the available 19 clinical isolates were included in this dataset; these isolates were so highly clonal that adding more genomes from the outbreak would not provide additional insights. This dataset also includes seven closely related outgroup isolates that were not part of the outbreak.

A S. enterica ser. Bareilly dataset (Table S4) was derived from a 2012 outbreak in mid-Atlantic US states associated with spicy tuna sushi rolls (CDC, 2012). Both epidemiological data and WGS data indicate that patients in the United States became infected with S. enterica ser. Bareilly by consuming tuna scrape that had been imported for making spicy tuna sushi from a fishery in India (Hoffmann et al., 2015). This benchmark dataset includes 18 clonal outbreak taxa, comprising both clinical and food isolates. Five outgroups are also included in this dataset, one of which was closed and serves as the reference genome.

The simulated dataset (Table S5) was generated from the empirical Salmonella phylogeny described above. This dataset is useful for validating the number and location of SNPs identified from a given bioinformatics pipeline and can help measure exactly how close an inferred phylogeny is to the true phylogeny since the “true” phylogeny is known in this case. This dataset comprises 18 simulated outbreak isolates and five outgroups, mirroring the empirical tree.

The dataset format

Tables 1 and 2 list the standardized descriptions used in each dataset, beginning with the required key/value pairs, followed by the available field names. Table 3 illustrates the use of this standardized reporting structure: columns in this format provide accession numbers for the sequence and phylogenetic tree data. Columns also contain epidemiological data characterizing the isolate as inside or outside of that specific outbreak. These data are housed at NCBI, a partner of the International Nucleotide Sequence Database Collaboration (INSDC) (Karsch-Mizrachi et al., 2012), and at OpenTree (Hinchliff et al., 2015). The tree topologies provided for each dataset (Fig. 1) were robust to different phylogenomic pipelines, such Lyve-Set (another SNP-based pipeline) (Katz et al., 2017) and wgMLST (allele-based pipeline) (Moura et al., 2016). To the best of our knowledge, the tree accompanying each dataset closely represents the true phylogeny, given the current taxon sampling and accepted epidemiology. For each benchmark dataset we include the following data:

Table 2 Metadata table body.

Fields included in the body of the metadata table that describe the individual sequences included in the dataset. The required fields are biosample_acc, strain, and sra_acc. Any optional field can be blank or contain a dash (−) if no value is given. Field names are case insensitive.

Field	Description	Required	Example value(s)	
biosample_acc	The identifier found in the NCBI BioSample database. This usually starts with SAMN or SAME.	Yes	SAMN01939119	
Strain	The name of the isolate	Yes	CFSAN002349	
genBankAssembly	The GenBank assembly identifier	No	GCA_001257675.1	
SRArun_acc	The Sequence Read Archive identifier	Yes	SRR1206159	
outbreak	If the isolate is associated with the outbreak or recall, list the PulseNet outbreak code, or other event identifier here.	No	1408MLGX6-3WGS outgroup	
datasetname	To which dataset this isolate belongs	Yes	1408MLGX6-3WGS	
suggestedReference	For reference-based pipelines, a dataset can suggest which reference assembly to use	Yes	TRUE
FALSE	
sha256sumAssembly	The sha256 checksum of the genome assembly. This will help assure that the download is successful.	Yes	9b926bc0adbea331a0a71f7bf18f6c7a62ebde7dd7a52fabe602ad8b00722c56	
sha256sumRead1	The sha256 checksum of the forward read	Yes	c43c41991ad8ed40ffcebbde36dc9011f471dea643fc8f715621a2e336095bf5	
sha256sumRead2	The sha256 checksum of the reverse read	Yes	4d12ed7e34b2456b8444dd71287cbb83b9c45bd18dc23627af0fbb6014ac0fca	

Table 3 Example dataset.

This as an example metadata table for a hypothetical single-isolate dataset, combining the header and body from Tables 1 and 2.

Organism	Listeria monocytogenes	
Outbreak	1408MLGX6-3WGS	
PMID	25789745	
Tree	http://api.opentreeoflife.org/v2/study/ot_301/tree/tree2.tre	
Source	Cheryl Tarr	
DataType	Empirical	
IntendedUse	Epi-validated outbreak	
biosample_acc	Strain	genBankAssembly	SRArun_acc	outbreak	datasetname	suggestedReference	sha256sumAssembly	sha256sumRead1	sha256sumRead2	
SAMN01939119	CFSAN002349	GCA_001257675.1	SRR1206159	1408MLGX6-3WGS	1408MLGX6-3WGS	TRUE	9b926bc0adbea331a0a71f7bf18f6c7a62ebde7dd7a52fabe602ad8b00722c56	c43c41991ad8ed40ffcebbde36dc9011f471dea643fc8f 715621a2e336095bf5	4d12ed7e34b2456b8444dd71287cbb83b9c 45bd18dc23627af0fbb6014ac0fca	

1. NCBI Sequence Read Archive (SRA) accessions for each isolate.

2. NCBI BioSample accession for each isolate.

3. A link to a maximum likelihood phylogenetic tree stored at the OpenTreeOfLife

4. NCBI assembly accessions for annotated draft and complete assemblies (where available). Information is provided about which assembly is appropriate for use as a reference.

The benchmark table format is a spreadsheet divided into two sections: a header and the body. The header contains generalized information of the dataset in a key/value format where column A is the key and the value is in column B. The available keys with example values are given in Table 1. Any property in the header applies to all genomes; for example, all isolates described in the spreadsheet should be of the same organism as listed in the header. The body of the dataset provides information for each taxon, or tip in the tree. Accessions, strain IDs, key to isolates in clonal event, and sha256sums are included here (Table 2). An example is given in Table 3.

Figure 1 The “true” phylogeny included for each dataset.

The outbreak or event-related taxa are colored red. (A) Listeria monocytogenes, (B) Escherichia coli, (C) Salmonella enterica, (D) Campylobacter jejuni, (E) simulated dataset.

Table 4 Benchmark datasets.

The key features of the four empirical and one simulated dataset are summarized in this table.

Dataset	Organism	Number of isolatesa	Epidemiologically linked isolatesb	Reference genomec	Type of dataset	Reference/Comment	
Stone Fruit Food recall	L. monocytogenes	31	28	CFSAN023463	Empirical	PMID: 27694232	
Spicy Tuna outbreak	S. enterica	23	18	CFSAN000189	Empirical	PMID: 25995194	
Raw Milk Outbreak	C. jejuni	22	14	D7331	Empirical	http://www.outbreakdatabase.com/details/hendricks-farm-and-dairy-raw-milk-2008/	
Sprouts Outbreak	E. coli	10	3	2011C-3609	Empirical	http://www.cdc.gov/ecoli/2014/o121-05-14/index.html	
Simulated outbreak	S. enterica	23	18	CFSAN000189	Synthetic	Simulated dataset based off the S. enterica spicy tuna outbreak tree and reference genome.	
Notes.

a Number of Isolates: total number of isolates in the dataset.

b Epidemiologically linked isolates: number of isolates implicated in the recall or outbreak.

c Reference genome: suggested reference genome for SNP analysis.

To ensure that every dataset is easily and reliably downloadable for anyone to use, we have created a script called Gen-FS Gopher (GG) that automates the download process. GG downloads the assemblies, raw reads, and tree(s) listed in a given dataset spreadsheet. Additionally, GG uses the sha256sum program to verify each download. Because some files depend on others (e.g., downloading the reverse read depends on the forward read; the sha256sha256 checksums depend on all reads being downloaded), GG creates a Makefile, which is then executed. That Makefile creates a dependency tree such that all files will be downloaded in the order they are needed. Each of our five benchmark datasets, described in Table 4, can be downloaded using this GG script.

Conclusion

The analysis and interpretation of datasets at the genomic scale is challenging due to the volume of data as well as the complexity and number of software programs often involved in the process. To have confidence in such analyses, it is important to be able to verify the performance of methods against datasets where the answers are already known. Ideally, such datasets provide a basis for not just testing methods, but also helping to provide a basis for ensuring the reproducibility of new methods and establishing comparability between bioinformatics pipelines. Having an established table format and tools to ensure easy and accurate downloads of benchmark datasets will help codify how data can be shared and evaluated. Here we have described five such datasets relevant for bacterial foodborne investigations based on WGS data. We have also established a standard file format suitable for these and future benchmark datasets, along with a script that is able to read and properly download them. It is to be emphasized that these benchmark datasets are useful for comparisons of phylogenomic pipelines and do not replace a more extensive validation of new pipelines. Such a new pipeline must be validated for typability, reproducibility, repeatability, discriminatory power, and epidemiological concordance using extensive isolate collections that are representative for the correct epidemiological context (Van Belkum et al., 2007).

The Gen-FS Gopher script along with five new benchmark datasets encourages reproducibility in the rapidly growing field of phylogenomics for pathogen surveillance. Currently, when new datasets are published the accessions to each data piece are embedded in a table within the body of the manuscript. Extracting these accessions from a PDF file can be arduous for large datasets. Without the GG script one would have to write their own program for downloading data from multiple databases (BioSample, SRA, GenBank, Assembly database at NCBI, and OpenTreeOfLife) or manually browse each database using cut/paste operations for each accession, downloading one by one. Using either route, the end result is often a directory of unorganized files and inconsistent file names, requiring tedious hand manipulation to get the correct file names and structure set up for local analysis. Because any given table of data is not in a standardized format, this process becomes a one-off, and the process has to be onerously reinvented for each table. Each step of this manual process increases the risk for error and degrades reproducibility. Our datasets and download script democratize this process: a single command can be cut/pasted into a unix/linux terminal, resulting in the automated download of the entire dataset (tree, raw fastq files, and assembly files) organized correctly for downstream analysis.

Further experimental validation of these and future empirical datasets will strengthen this resource. We will continue to work on these datasets using Sanger-sequence validation and will encourage future submitters to validate their datasets, too. Additionally, we encourage future submitters to make their entire datasets available through INSDC and OpenTree in our recommended format. The participants in Gen-FS are also starting a collaboration with the Global Microbial Identifier Program (Global Microbial Identifier, 2011) that goes beyond the annual GMI Proficiency Test. Researchers from around the world will be encouraged to contribute validated empirical and simulated datasets, providing a more diverse set of benchmark datasets. To aid in quality assurance, we suggest a minimum of 20× coverage for each genome in a dataset. Submissions following our described spreadsheet format will ensure compatibility with our download script, and should include isolates with as much BioSample metadata as possible including values such as the outbreak code and isolate source (e.g., clinical or food/environmental). Our work will allow other researchers to contribute benchmark datasets for testing and comparing bioinformatics pipelines, which will contribute to more robust and reliable analyses of genomic diversity. The GitHub page for that effort can be accessed here: https://github.com/globalmicrobialidentifier-WG3/datasets.

Supplemental Information

Table S1 Listeria monocytogenes stone fruit food recall dataset

This standard dataset table can be used for easy download of linked data using the Gen-FS Gopher script described in this manuscript. Save as a tsv file to run the script.

Click here for additional data file.

Table S2 Campylobacter jejun i raw milk outbreak dataset

This standard dataset table can be used for easy download of linked data using the Gen-FS Gopher script described in this manuscript. Save as a tsv file to run the script.

Click here for additional data file.

Table S3 Escherichia coli sprouts outbreak dataset

This standard dataset table can be used for easy download of linked data using the Gen-FS Gopher script described in this manuscript. Save as a tsv file to run the script.

Click here for additional data file.

Table S4 Salmonella enterica epi-validated outbreak dataset

This standard dataset table can be used for easy download of linked data using the Gen-FS Gopher script described in this manuscript. Save as a tsv file to run the script.

Click here for additional data file.

Table S5 Simulated outbreak from known phylogeny and reference genome

This standard dataset table can be used for easy download of linked data using the Gen-FS Gopher script described in this manuscript. Save as a tsv file to run the script.

Click here for additional data file.

We would like to thank Chris Tillman at CFSAN and Cheryl Tarr at CDC for sequencing work on L. monocytogenes. We would also like to thank Collette Fitzgerald, Vikrant Dutta, Janet Pruckler, and Grant Williams from CDC in helping identify and sequence the isolates from a well understood Campylobacter outbreak. Additionally, Andre Weltman and Lisa Dettinger from the Pennsylvania Department of Health gave vital information pertaining to the Campylobacter outbreak. We would like to acknowledge Philip Bronstein at FSIS-USDA for his efforts. Lastly, we would like to acknowledge Lili Fox Vélez from FDA for scientific writing support.

Additional Information and Declarations

Competing Interests

Author Contributions

DNA Deposition

Data Availability

The authors declare there are no competing interests.

Ruth E. Timme and Lee S. Katz conceived and designed the experiments, performed the experiments, analyzed the data, wrote the paper, prepared figures and/or tables, reviewed drafts of the paper.

Hugh Rand conceived and designed the experiments, contributed reagents/materials/analysis tools, reviewed drafts of the paper.

Martin Shumway, Eija K. Trees, Mustafa Simmons, Richa Agarwala, Steven Davis, Glenn E. Tillman, Stephanie Defibaugh-Chavez, Heather A. Carleton and William A. Klimke contributed reagents/materials/analysis tools, reviewed drafts of the paper.

The following information was supplied regarding the deposition of DNA sequences:

NCBI accessions (SRA, Biosample, Assembly, etc.) are provided in the Supplemental Files.

The following information was supplied regarding data availability:

GitHub: https://github.com/WGS-standards-and-analysis/datasets.

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
