# Peer review of "Benchmark datasets for phylogenomic pipeline validation, applications for foodborne pathogen surveillance"

_PeerJ, doi:10.7717/peerj.3893_

## Round 0.1 · original submission · Minor Revisions

I have now received three reviews of your paper and all reviewers found your paper interesting, well-done, and appropriate for the journal. All reviewers, however, have outlined a number of issues for you to deal with in a revision. Please especially pay close attention to the comment from two reviewers about the non-trivial disconnect between the .xlsx example data and the .tsv requirement of the pipeline. This is problematic. Reviewers also noted the need for standardized datasets and improved documentation. I hope you will find these reviews helpful in your revision. Please be sure to develop a cover letter that addresses well each point from each reviewer. Thanks. Good luck with the revision.

·

Basic reporting

There's no question about basic reporting in this manuscript. Professional English is used throughout, sufficient background is provided, and the structure is professional and the manuscript self-contained

Experimental design

Manuscript adheres to Aims and Scope. The research question is well defined and relevant, and the investigation has been performed to high technical and ethical standards.

Validity of the findings

The findings are valid and the manuscript adheres to PeerJ's standards

Additional comments

I think it’ll be helpful if the authors add information as to how they determined that the bacterial isolates are epidemiologically linked and in what direction. One key issue is establishing a phylogenetic relationship among isolates that mirrors the transmission network. I understand this is straightforward for the simulated data but it’s not clear to me how this was done for the real biological data.

I mention this because the equivalency of phylogenies to transmission trees has been somewhat controversial (e.g., PMID: 24037268; PMID: 24675511; PMID: 26230489; PMID: 24916411). Consequently, methods has been develop to directly address this issue, e.g., PMID: 28545083

One minor thing that's hanging in my head is why providing example datasets in xlsx if the script takes tsv’s? The conversion is trivial but keeping everything tidy in open formats has its advantages.

·

Basic reporting

Overall, the manuscript was well written. It is unclear why so many references are applied to the reference of Lyve-SET. I would just cite the manuscript that describes the pipeline, unless you are citing the specifics of diverse projects where Lyve-SET has been used. The NASP pipeline was mentioned, but wasn’t cited.

L95: Peptoclostridium is not really used and is immediately followed by Clostridium in your text. GenBank currently uses “Clostrioides”.

Experimental design

This manuscript describes a method to download and compare pipelines for WGS analyses. However, after downloading the repository, I saw no script referred to as “Gen-FS Gopher” or “GG”. There appears to be no comprehensive, documented workflow on how to download the 5 test datasets. The scripts provided to convert between “xlsx” and “tsv” required extra dependencies and I could not get them to work. Saving them as tsv files seemed to work and I was able to download the datasets, although I received an error.

If these are benchmarking datasets, which SNPs should consistently be called? Simply obtaining the same tree doesn’t mean that any given pipeline is calling the correct SNPs. Why not release the SNPs that were used to infer the provided tree? As it reads now, it is unclear if the true utility of this paper is in benchmarking, or providing a tool to interface with the SRA to grab sequences and trees.

Validity of the findings

Having a standard benchmarking scheme would be very helpful to standardize analyses across computational pipelines. Additional work on the repository associated with testing your scripts on multiple platforms and adding additional documentation would help users not familiar with these types of pipelines. Additional text on how these datasets can be used to benchmark pipelines would also help users assess new pipelines, using metrics that can be directly compared.

Additional comments

This manuscript could be improved by working on the documentation, making sure that script names in the manuscript match those in the repository, and by providing specific metrics that users could use to assess new SNP pipelines. Providing standardized datasets to the community would be very helpful.

·

Basic reporting

This article is generally clear and well written and the impact of the work is communicated clearly. Data and programs were accessible as indicated in the text.

I found that the text was not clearly organized into Introduction/Materials and Methods, Results and Discussion. For example, the last paragraph of the introduction seems to be materials and methods. Some of the information on lines 102-112 (e.g. selection of datasets, data simulation) should be integrated into the materials and methods. Similarly, description of data simulation (lines 148-160) and the description of the development of the GG script also seem to fit within the scope of materials and methods. Authors should consider modifying organization and perhaps combine results/discussion. Given that the topic of this paper is somewhat different than a traditional scientific paper (announcement of the availability of data/tools rather than an evaluation of performance), perhaps traditional subheadings are not appropriate.

Some minor comments:
The Gen-FS gopher script is called “downloadDataset” on the github site which may lead to confusion.

The pages preceding the tables are confusing. Eg:Table 2 “ Reviews and evaluates data submissions in food and color additive petitions and premarket notifications (GRAS and Food Contact Surfaces notifications) to determine the safety of the use of a product in
foods within the context of applicab”

Line 27: ortholog determination (no hyphen)
Line 28: single-nucleotide polymorphisms
Line 60/71: publically (should be publicly as in line 37?)
Line 73: update number of genomes if possible

Experimental design

The design of the study is clearly described.

I would be interested in results of analysis of this dataset with different phylogenetic pipelines that are currently in use among Gen FS partners to demonstrate how data generated from this dataset would be interpreted.

Given that reference genomes are provided, could SNP/SNVs locations be identified?

It would also be appropriate to provide a figure showing phylogenetic trees.

Validity of the findings

no comment

Additional comments

This manuscript describes the development of benchmark NGS datasets for priority foodborne pathogens that can be used to evaluate the performance of phylogenomic pipelines, and addresses the development of standardized formats for submission of similar benchmark datasets in the future. This work represents an important contribution that will be particularly useful for regulators in determining the reliability of new phylogenomic pipelines applied to the surveillance of foodborne pathogens. The tool for rapid retrieval of validation data will be useful in providing easy access to this data.

---

## Round 0.2 · accepted · Accept

Thank you for your careful attention to the previous reviews. I feel you have accommodated the reviewers' concern well and that your paper is now ready for acceptance. Congratulations and thanks for submitting to PeerJ.